# On the Accuracy of Self-Normalized Log-Linear Models

**Jacob Andreas**,* **Maxim Rabinovich**,* **Michael I. Jordan**, **Dan Klein**
Computer Science Division, University of California, Berkeley
{jda,rabinovich,jordan,klein}@cs.berkeley.edu

## Abstract

Calculation of the log-normalizer is a major computational obstacle in applications of log-linear models with large output spaces. The problem of fast normalizer computation has therefore attracted significant attention in the theoretical and applied machine learning literature. In this paper, we analyze a recently proposed technique known as "self-normalization", which introduces a regularization term in training to penalize log normalizers for deviating from zero. This makes it possible to use unnormalized model scores as approximate probabilities. Empirical evidence suggests that self-normalization is extremely effective, but a theoretical understanding of why it should work, and how generally it can be applied, is largely lacking.

We prove upper bounds on the loss in accuracy due to self-normalization, describe classes of input distributions that self-normalize easily, and construct explicit examples of high-variance input distributions. Our theoretical results make predictions about the difficulty of fitting self-normalized models to several classes of distributions, and we conclude with empirical validation of these predictions.

## 1 Introduction

Log-linear models, a general class that includes conditional random fields (CRFs) and generalized linear models (GLMs), offer a flexible yet tractable approach modeling conditional probability distributions $p(x|y)$ [1, 2]. When the set of possible $y$ values is large, however, the computational cost of computing a normalizing constant for each $x$ can be prohibitive—involving a summation with many terms, a high-dimensional integral or an expensive dynamic program.

The machine translation community has recently described several procedures for training "self-normalized" log-linear models [3, 4]. The goal of self-normalization is to choose model parameters that simultaneously yield accurate predictions and produce normalizers clustered around unity. Model scores can then be used as approximate surrogates for probabilities, obviating the computation normalizer computation.

In particular, given a model of the form

$$p_\eta(y \mid x) = e^{\eta^T T(y,\, x) - A(\eta,\, x)} \tag{1}$$

with

$$A(\eta,\, x) = \log \sum_{y \in \mathcal{Y}} e^{\eta^T T(y,\, x)} \,, \tag{2}$$

we seek a setting of $\eta$ such that $A(x, \eta)$ is close enough to zero (with high probability under $p(x)$) to be ignored.

---

This paper aims to understand the theoretical properties of self-normalization. Empirical results have already demonstrated the efficacy of this approach—for discrete models with many output classes, it appears that normalizer values can be made nearly constant without sacrificing too much predictive accuracy, providing dramatic efficiency increases at minimal performance cost.

The broad applicability of self-normalization makes it likely to spread to other large-scale applications of log-linear models, including structured prediction (with combinatorially many output classes) and regression (with continuous output spaces). But it is not obvious that we should expect such approaches to be successful: the number of inputs (if finite) can be on the order of millions, the geometry of the resulting input vectors $x$ highly complex, and the class of functions $A(\eta, x)$ associated with different inputs quite rich. To find to find a nontrivial parameter setting with $A(\eta, x)$ roughly constant seems challenging enough; to require that the corresponding $\eta$ also lead to good classification results seems too much. And yet for many input distributions that arise in practice, it appears possible to choose $\eta$ to make $A(\eta, x)$ nearly constant without having to sacrifice classification accuracy.

Our goal is to bridge the gap between theoretical intuition and practical experience. Previous work [5] bounds the sample complexity of self-normalizing training procedures for a restricted class of models, but leaves open the question of how self-normalization interacts with the predictive power of the learned model. This paper seeks to answer that question. We begin by generalizing the previously-studied model to a much more general class of distributions, including distributions with continuous support (Section 3). Next, we provide what we believe to be the first characterization of the interaction between self-normalization and model accuracy Section 4. This characterization is given from two perspectives:

- a bound on the "likelihood gap" between self-normalized and unconstrained models
- a conditional distribution provably hard to represent with a self-normalized model

In Figure 5, we present empirical evidence that these bounds correctly characterize the difficulty of self-normalization, and in the conclusion we survey a set of open problems that we believe merit further investigation.

## 2  Problem background

The immediate motivation for this work is a procedure proposed to speed up decoding in a machine translation system with a neural-network language model [3]. The language model used is a standard feed-forward neural network, with a "softmax" output layer that turns the network's predictions into a distribution over the vocabulary, where each probability is log-proportional to its output activation. It is observed that with a sufficiently large vocabulary, it becomes prohibitive to obtain probabilities from this model (which must be queried millions of times during decoding). To fix this, the language model is trained with the following objective:

$$\max_W \sum_i \left[ N(y_i|x_i; W) - \log \sum_{y'} e^{N(y'|x_i; W)} - \alpha \left( \log \sum_{y'} e^{N(y'|x_i; W)} \right)^2 \right]$$

where $N(y|x; W)$ is the response of output $y$ in the neural net with weights $W$ given an input $x$. From a Lagrangian perspective, the extra penalty term simply confines the $W$ to the set of "empirically normalizing" parameters, for which all log-normalizers are close (in squared error) to the origin. For a suitable choice of $\alpha$, it is observed that the trained network is simultaneously accurate enough to produce good translations, and close enough to self-normalized that the raw scores $N(y_i|x_i)$ can be used in place of log-probabilities without substantial further degradation in quality.

We seek to understand the observed success of these models in finding accurate, normalizing parameter settings. While it is possible to derive bounds of the kind we are interested in for general neural networks [6], in this paper we work with a simpler linear parameterization that we believe captures the interesting aspects of this problem. [1]

**Related work**

The approach described at the beginning of this section is closely related to an alternative self-normalization trick described based on noise-contrastive estimation (NCE) [8]. NCE is an alternative to direct optimization of likelihood, instead training a classifier to distinguish between true samples from the model, and "noise" samples from some other distribution. The structure of the training objective makes it possible to replace explicit computation of each log-normalizer with an estimate. In traditional NCE, these values are treated as part of the parameter space, and estimated simultaneously with the model parameters; there exist guarantees that the normalizer estimates will eventually converge to their true values. It is instead possible to fix all of these estimates to one. In this case, empirical evidence suggests that the resulting model will also exhibit self-normalizing behavior [4].

A host of other techniques exist for solving the computational problem posed by the log-normalizer. Many of these involve approximating the associated sum or integral using quadrature [9], herding [10], or Monte Carlo methods [11]. For the special case of discrete, finite output spaces, an alternative approach—the hierarchical softmax—is to replace the large sum in the normalizer with a series of binary decisions [12]. The output classes are arranged in a binary tree, and the probability of generating a particular output is the product of probabilities along the edges leading to it. This reduces the cost of computing the normalizer from $\mathcal{O}(k)$ to $\mathcal{O}(\log k)$. While this limits the set of distributions that can be learned, and still requires greater-than-constant time to compute normalizers, it appears to work well in practice. It cannot, however, be applied to problems with continuous output spaces.

## 3 Self-normalizable distributions

We begin by providing a slightly more formal characterization of a general log-linear model:

**Definition 1 (Log-linear models).** Given a space of *inputs* $\mathcal{X}$, a space of *outputs* $\mathcal{Y}$, a measure $\mu$ on $\mathcal{Y}$, a nonnegative function $h : \mathcal{Y} \to \mathbb{R}$, and a function $T : \mathcal{X} \times \mathcal{Y} \to \mathbb{R}^d$ that is $\mu$-measurable with respect to its second argument, we can define a *log-linear model* indexed by parameters $\eta \in \mathbb{R}^d$, with the form

$$p_\eta(y|x) = h(y)e^{\eta^\top T(x,y) - A(x,\eta)} \,, \tag{3}$$

where

$$A(x,\eta) \triangleq \log \int_{\mathcal{Y}} h(y)e^{\eta^\top T(x,y)} \, \mathrm{d}\mu(y) \,. \tag{4}$$

If $A(x,\eta) \leq \infty$, then $\int_y p_\eta(y|x) \, \mathrm{d}\mu(y) = 1$, and $p_\eta(y|x)$ is a probability density over $\mathcal{Y}$.[2]

We next formalize our notion of a self-normalized model.

**Definition 2 (Self-normalized models).** The log-linear model $p_\eta(y|x)$ is *self-normalized with respect to* a set $\mathcal{S} \subset \mathcal{X}$ if for all $x \in \mathcal{S}$, $A(x,\eta) = 0$. In this case we say that $\mathcal{S}$ is *self-normalizable*, and $\eta$ is *self-normalizing* w.r.t. $\mathcal{S}$.

An example of a normalizable set is shown in Figure 1a, and we provide additional examples below:

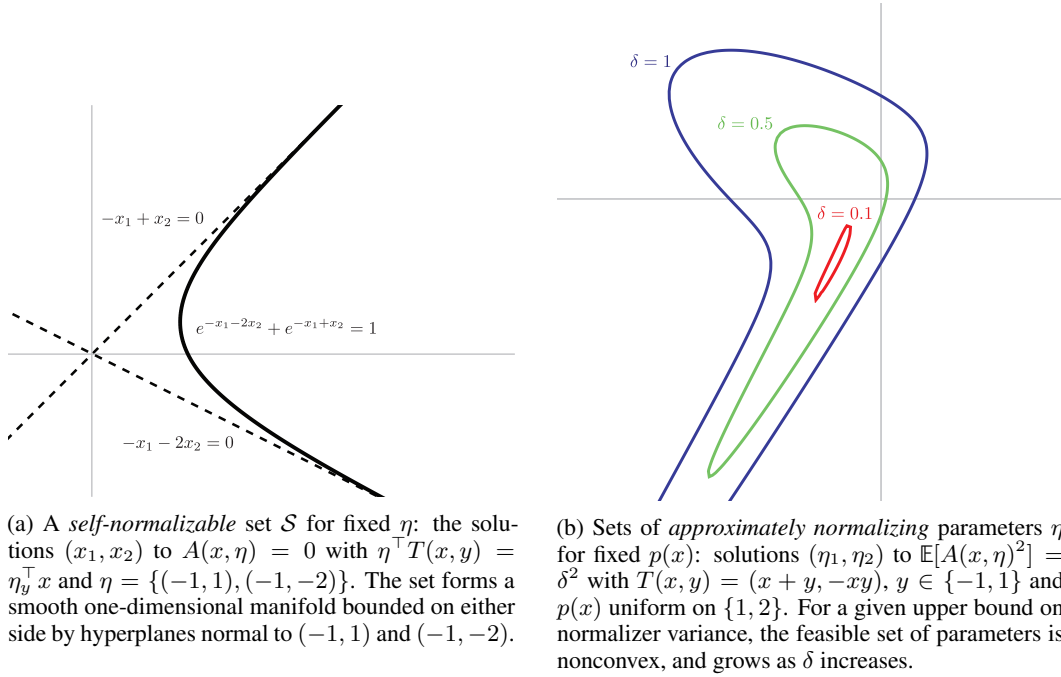

(a) A *self-normalizable* set $\mathcal{S}$ for fixed $\eta$: the solutions $(x_1, x_2)$ to $A(x, \eta) = 0$ with $\eta^\top T(x, y) = \eta_y^\top x$ and $\eta = \{(-1, 1), (-1, -2)\}$. The set forms a smooth one-dimensional manifold bounded on either side by hyperplanes normal to $(-1, 1)$ and $(-1, -2)$.

(b) Sets of *approximately normalizing* parameters $\eta$ for fixed $p(x)$: solutions $(\eta_1, \eta_2)$ to $\mathbb{E}[A(x, \eta)^2] = \delta^2$ with $T(x, y) = (x + y, -xy)$, $y \in \{-1, 1\}$ and $p(x)$ uniform on $\{1, 2\}$. For a given upper bound on normalizer variance, the feasible set of parameters is nonconvex, and grows as $\delta$ increases.

Figure 1: Self-normalizable data distributions and parameter sets.

**Example.** *Suppose*

$$\mathcal{S} = \{\log 2, -\log 2\},$$
$$\mathcal{Y} = \{-1, 1\}$$
$$T(x, y) = [xy, 1]$$
$$\eta = (1, \log(2/5)).$$

*Then for either $x \in \mathcal{S}$,*

$$A(x, \eta) = \log(e^{\log 2 + \log(2/5)} + e^{-\log 2 + \log(2/5)})$$
$$= \log((2/5)(2 + 1/2))$$
$$= 0,$$

*and $\eta$ is self-normalizing with respect to $\mathcal{S}$.*

It is also easy to choose parameters that do not result in a self-normalized distribution, and in fact to construct a target distribution which cannot be self-normalized:

**Example.** *Suppose*

$$\mathcal{X} = \{(1, 0), (0, 1), (1, 1)\}$$
$$\mathcal{Y} = \{-1, 1\}$$
$$T(x, y) = (x_1 y, x_2 y, 1)$$

*Then there is no $\eta$ such that $A(x, \eta) = 0$ for all $x$, and $A(x, \eta)$ is constant if and only if $\eta = \mathbf{0}$.*

As previously motivated, downstream uses of these models may be robust to small errors resulting from improper normalization, so it would be useful to generalize this definition of normalizable distributions to distributions that are only approximately normalizable. Exact normalizability of the conditional distribution is a deterministic statement—there either does or does not exist some $x$ that violates the constraint. In Figure 1a, for example, it suffices to have a single $x$ off of the indicated surface to make a set non-normalizable. Approximate normalizability, by contrast, is inherently a *probabilistic* statement, involving a distribution $p(x)$ over inputs. Note carefully that we are attempting to represent $p(y|x)$ but have no representation of (or control over) $p(x)$, and that approximate normalizability depends on $p(x)$ but not $p(y|x)$.

Informally, if some input violates the self-normalization constraint by a large margin, but occurs only very infrequently, there is no problem; instead we are concerned with *expected* deviation. It

is also at this stage that the distinction between penalization of the normalizer vs. log-normalizer becomes important. The normalizer is necessarily bounded below by zero (so overestimates might appear much worse than underestimates), while the log-normalizer is unbounded in both directions. For most applications we are concerned with log probabilities and log-odds ratios, for which an expected normalizer close to zero is just as bad as one close to infinity. Thus the log-normalizer is the natural choice of quantity to penalize.

**Definition 3** (**Approximately self-normalized models**). The log-linear distribution $p_\eta(y|x)$ is $\delta$-*approximately normalized with respect to* a distribution $p(x)$ over $\mathcal{X}$ if $\mathbb{E}[A(X,\eta)^2] < \delta^2$. In this case we say that $p(x)$ is $\delta$-*approximately self-normalizable*, and $\eta$ is $\delta$-*approximately self-normalizing*.

The sets of $\delta$-approximately self-normalizing parameters for a fixed input distribution and feature function are depicted in Figure 1b. Unlike self-normalizable sets of *inputs*, self-normalizing and approximately self-normalizing sets of *parameters* may have complex geometry.

Throughout this paper, we will assume that vectors of sufficient statistics $T(x,y)$ have bounded $\ell_2$ norm at most $R$, natural parameter vectors $\eta$ have $\ell_2$ norm at most $B$ (that is, they are Ivanov-regularized), and that vectors of both kinds lie in $\mathbb{R}^d$. Finally, we assume that all input vectors have a constant feature—in particular, that $x_0 = 1$ for every $x$ (with corresponding weight $\eta_0$). [3]

The first question we must answer is whether the problem of training self-normalized models is feasible at all—that is, whether there exist any exactly self-normalizable data distributions $p(x)$, or at least $\delta$-approximately self-normalizable distributions for small $\delta$. Section 3 already gave an example of an exactly normalizable distribution. In fact, there are large classes of both exactly and approximately normalizable distributions.

**Observation.** *Given some fixed $\eta$, consider the set $S_\eta = \{x \in \mathcal{X} : A(x,\eta) = 0\}$. Any distribution $p(x)$ supported on $S_\eta$ is normalizable. Additionally, every self-normalizable distribution is characterized by at least one such $\eta$.*

This definition provides a simple geometric characterization of self-normalizable distributions. An example solution set is shown in Figure 1a. More generally, if $y$ is discrete and $T(x,y)$ consists of $|\mathcal{Y}|$ repetitions of a fixed feature function $t(x)$ (as in Figure 1a), then we can write

$$A(x,\eta) = \log \sum_{y \in \mathcal{Y}} e^{\eta_y^\top t(x)}. \tag{5}$$

Provided $\eta_y^\top t(x)$ is convex in $x$ for each $\eta_y$, the level sets of $A$ as a function of $x$ form the boundaries of convex sets. In particular, exactly normalizable sets are always the boundaries of convex regions, as in the simple example Figure 1a.

We do not, in general, expect real-world datasets to be supported on the precise class of self-normalizable surfaces. Nevertheless, it is very often observed that data of practical interest lie on other low-dimensional manifolds within their embedding feature spaces. Thus we can ask whether it is sufficient for a target distribution to be well-approximated by a self-normalizing one. We begin by constructing an appropriate measurement of the quality of this approximation.

**Definition 4** (**Closeness**). An input distribution $p(x)$ is $D$-*close* to a set $\mathcal{S}$ if

$$\mathbb{E}\left[\inf_{x^* \in \mathcal{S}} \sup_{y \in \mathcal{Y}} ||T(X,y) - T(x^*,y)||_2\right] \leq D \tag{6}$$

In other words, $p(x)$ is $D$-close to $\mathcal{S}$ if a random sample from $p$ is no more than a distance $D$ from $\mathcal{S}$ in expectation. Now we can relate the quality of this approximation to the level of self-normalization achieved. Generalizing a result from [5], we have:

**Proposition 1.** *Suppose $p(x)$ is $D$-close to $\{x : A(x,\eta) = 1\}$. Then $p(x)$ is $BD$-approximately self-normalizable (recalling that $||x||_2 \leq B$).*

(Proofs for this section may be found in Appendix A.)

The intuition here is that data distributions that place most of their mass in feature space close to normalizable sets are approximately normalizable on the same scale.

## 4 Normalization and model accuracy

So far our discussion has concerned the problem of finding conditional distributions that self-normalize, without any concern for how well they actually perform at modeling the data. Here the relationship between the approximately self-normalized distribution and the true distribution $p(y|x)$ (which we have so far ignored) is essential. Indeed, if we are not concerned with making a good model it is always trivial to make a normalized one—simply take $\eta = \mathbf{0}$ and then scale $\eta_0$ appropriately! We ultimately desire both good self-normalization and good data likelihood, and in this section we characterize the tradeoff between maximizing data likelihood and satisfying a self-normalization constraint.

We achieve this characterization by measuring the *likelihood gap* between the classical maximum likelihood estimator, and the MLE subject to a self-normalization constraint. Specifically, given pairs $((x_1, y_1), (x_2, y_2), \dots, (x_n, y_n))$, let $\ell(\eta|x, y) = \sum_i \log p_\eta(y_i|x_i)$. Then define

$$\hat{\eta} = \arg \max_\eta \ell(\eta|x, y) \tag{7}$$

$$\hat{\eta}_\delta = \arg \max_{\eta : V(\eta) \leq \delta} \ell(\eta|x, y) \tag{8}$$

(where $V(\eta) = \frac{1}{n} \sum_i A(x_i, \eta)^2$).

We would like to obtain a bound on the *likelihood gap*, which we define as the quantity

$$\Delta_\ell(\hat{\eta}, \hat{\eta}_\delta) = \frac{1}{n} (\ell(\hat{\eta}|x, y) - \ell(\hat{\eta}_\delta|x, y)) . \tag{9}$$

We claim:

**Theorem 2.** *Suppose $\mathcal{Y}$ has finite measure. Then asymptotically as $n \to \infty$*

$$\Delta_\ell(\hat{\eta}, \hat{\eta}_\delta) \leq \left( 1 - \frac{\delta}{R||\hat{\eta}||_2} \right) \mathbb{E} \, \mathrm{KL}(p_\eta(\cdot|X) \, || \, \mathrm{Unif}) . \tag{10}$$

(Proofs for this section may be found in Appendix B.)

This result lower-bounds the likelihood at $\hat{\eta}_\delta$ by explicitly constructing a scaled version of $\hat{\eta}$ that satisfies the self-normalization constraint. Specifically, if $\eta$ is chosen so that normalizers are penalized for distance from $\log \mu(\mathcal{Y})$ (e.g. the logarithm of the number of classes in the finite case), then any increase in $\eta$ along the span of the data is guaranteed to increase the penalty. From here it is possible to choose an $\alpha \in (0, 1)$ such that $\alpha \hat{\eta}$ satisfies the constraint. The likelihood at $\alpha \hat{\eta}$ is necessarily less than $\ell(\hat{\eta}_\delta|x, y)$, and can be used to obtain the desired lower bound.

Thus at one extreme, distributions close to uniform can be self-normalized with little loss of likelihood. What about the other extreme—distributions "as far from uniform as possible"? With suitable assumptions about the form of $p_{\hat{\eta}}(y|x)$, we can use the same construction of a self-normalizing parameter to achieve an alternative characterization for distributions that are close to deterministic:

**Proposition 3.** *Suppose that $\mathcal{X}$ is a subset of the Boolean hypercube, $\mathcal{Y}$ is finite, and $T(x, y)$ is the conjunction of each element of $x$ with an indicator on the output class. Suppose additionally that in* every *input $x$, $p_{\hat{\eta}}(y|x)$ makes a unique best prediction—that is, for each $x \in \mathcal{X}$, there exists a unique $y^* \in \mathcal{Y}$ such that whenever $y \neq y^*$, $\eta^\top T(x, y^*) > \eta^\top T(x, y)$. Then*

$$\Delta_\ell(\hat{\eta}, \hat{\eta}_\delta) \leq b \left( ||\eta||_2 - \frac{\delta}{R} \right)^2 e^{-c\delta/R} \tag{11}$$

*for distribution-dependent constants $b$ and $c$.*

This result is obtained by representing the constrained likelihood with a second-order Taylor expansion about the true MLE. All terms in the likelihood gap vanish except for the remainder; this can be

upper-bounded by the $||\hat\eta_\delta||_2^2$ times the largest eigenvalue the feature covariance matrix at $\hat\eta_\delta$, which in turn is bounded by $e^{-c\delta/R}$.

The favorable rate we obtain for this case indicates that "all-nonuniform" distributions are also an easy class for self-normalization. Together with Theorem 2, this suggests that hard distributions must have some mixture of uniform and nonuniform predictions for different inputs. This is supported by the results in Section 4.

The next question is whether there is a corresponding lower bound; that is, whether there exist any conditional distributions for which all nearby distributions are provably hard to self-normalize. The existence of a direct analog of Theorem 2 remains an open problem, but we make progress by developing a general framework for analyzing normalizer variance.

One key issue is that while likelihoods are invariant to certain changes in the natural parameters, the log normalizers (and therefore their variance) is far from invariant. We therefore focus on equivalence classes of natural parameters, as defined below. Throughout, we will assume a fixed distribution $p(x)$ on the inputs $x$.

**Definition 5** (**Equivalence of parameterizations**). Two natural parameter values $\eta$ and $\eta'$ are said to be *equivalent* (with respect to an input distribution $p(x)$), denoted $\eta \sim \eta'$ if

$$p_\eta(y|X) = p_{\eta'}(y|X) \quad \text{a.s. } p(x)$$

We can then define the optimal log normalizer variance for the distribution associated with a natural parameter value.

**Definition 6** (**Optimal variance**). We define the *optimal log normalizer variance* of the log-linear model associated with a natural parameter value $\eta$ by

$$V^*(\eta) = \inf_{\eta' \sim \eta} \text{Var}_{p(x)}\left[A(X,\ \eta)\right].$$

We now specialize to the case where $\mathcal{Y}$ is finite with $|\mathcal{Y}| = K$ and where $T : \mathcal{Y} \times \mathcal{X} \to \mathbb{R}^{Kd}$ satisfies

$$T(k,\ x)_{k'j} = \delta_{kk'}x_j.$$

This is an important special case that arises, for example, in multi-way logistic regression. In this setting, we can show that despite the fundamental non-identifiability of the model, the variance can still be shown to be high under *any* parameterization of the distribution.

**Theorem 4.** *Let $\mathcal{X} = \{0,\ 1\}^d$ and let the input distribution $p(x)$ be uniform on $\mathcal{X}$. There exists an $\eta^0 \in \mathbb{R}^{Kd}$ such that for $\eta = \alpha\eta^0$, $\alpha > 0$,*

$$V^*(\eta) \geq \frac{||\eta||_2^2}{32d(d-1)} - 4Ke^{-\frac{\sqrt{1-\frac{1}{d}}||\eta||_2}{2(d-1)}}||\eta||_2.$$

## 5 Experiments

The high-level intuition behind the results in the preceding section can be summarized as follows: 1) for predictive distributions that are in expectation high-entropy or low-entropy, self-normalization results in a relatively small likelihood gap; 2) for mixtures of high- and low-entropy distributions, self-normalization may result in a large likelihood gap. More generally, we expect that an increased tolerance for normalizer variance will be associated with a decreased likelihood gap.

In this section we provide experimental confirmation of these predictions. We begin by generating a set of random sparse feature vectors, and an initial weight vector $\eta_0$. In order to produce a sequence of label distributions that smoothly interpolate between low-entropy and high-entropy, we introduce a temperature parameter $\tau$, and for various settings of $\tau$ draw labels from $p_{\tau\eta}$. We then fit a self-normalized model to these training pairs. In addition to the synthetic data, we compare our results to empirical data [3] from a self-normalized language model.

Figure 2a plots the tradeoff between the likelihood gap and the error in the normalizer, under various distributions (characterized by their KL from uniform). Here the tradeoff between self-normalization and model accuracy can be seen—as the normalization constraint is relaxed, the likelihood gap decreases.

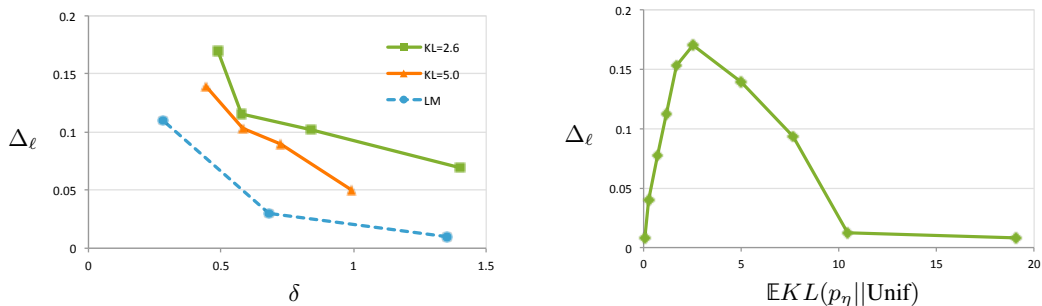

(a) Normalization / likelihood tradeoff. As the normalization constraint $\delta$ is relaxed, the likelihood gap $\Delta_\ell$ decreases. Lines marked "KL=" are from synthetic data; the line marked "LM" is from [3].

(b) Likelihood gap as a function of expected divergence from the uniform distribution. As predicted by theory, the likelihood gap increases, then decreases, as predictive distributions become more peaked.

Figure 2: Experimental results

Figure 2b shows how the likelihood gap varies as a function of the quantity $\mathbb{E}KL(p_\eta(\cdot|X)||\mathrm{Unif})$. As predicted, it can be seen that both extremes of this quantity result in small likelihood gaps, while intermediate values result in large likelihood gaps.

## 6 Conclusions

Motivated by the empirical success of self-normalizing parameter estimation procedures, we have attempted to establish a theoretical basis for the understanding of such procedures. We have characterized both self-normalizable *distributions*, by constructing provably easy examples, and *training procedures*, by bounding the loss of likelihood associated with self-normalization.

While we have addressed many of the important first-line theoretical questions around self-normalization, this study of the problem is by no means complete. We hope this family of problems will attract further study in the larger machine learning community; toward that end, we provide the following list of open questions:

1. **How else can the approximately self-normalizable distributions be characterized?** The class of approximately normalizable distributions we have described is unlikely to correspond perfectly to real-world data. We expect that Proposition 1 can be generalized to other parametric classes, and relaxed to accommodate spectral or sparsity conditions.

2. **Are the upper bounds in Theorem 2 or Proposition 3 tight?** Our constructions involve relating the normalization constraint to the $\ell_2$ norm of $\eta$, but in general some parameters can have very large norm and still give rise to almost-normalized distributions.

3. **Do corresponding lower bounds exist?** While it is easy to construct of exactly self-normalizable distributions (which suffer no loss of likelihood), we have empirical evidence that hard distributions also exist. It would be useful to lower-bound the loss of likelihood in terms of some simple property of the target distribution.

4. **Is the hard distribution in Theorem 4 stable?** This is related to the previous question. The existence of high-variance distributions is less worrisome if such distributions are fairly rare. If the lower bound falls off quickly as the given construction is perturbed, then the associated distribution may still be approximately self-normalizable with a good rate.

We have already seen that new theoretical insights in this domain can translate directly into practical applications. Thus, in addition to their inherent theoretical interest, answers to each of these questions might be applied directly to the training of approximately self-normalized models in practice. We expect that self-normalization will find increasingly many applications, and we hope the results in this paper provide a first step toward a complete theoretical and empirical understanding of self-normalization in log-linear models.

**Acknowledgments** The authors would like to thank Robert Nishihara for useful discussions. JA and MR are supported by NSF Graduate Fellowships, and MR is additionally supported by the Fannie and John Hertz Foundation Fellowship.

## Footnotes

[1]It is possible to view a log-linear model as a single-layer network with a softmax output. More usefully, all of the results presented here apply directly to trained neural nets in which the last layer only is *retrained* to self-normalize [7].

[2]Some readers may be more familiar with generalized linear models, which also describe exponential family distributions with a linear dependence on input. The presentation here is strictly more general, and has a few notational advantages: it makes explicit the dependence of $A$ on $x$ and $\eta$ but not $y$, and lets us avoid tedious bookkeeping involving natural and mean parameterizations. [13]

[3] It will occasionally be instructive to consider the special case where $\mathcal{X}$ is the Boolean hypercube, and we will explicitly note where this assumption is made. Otherwise all results apply to general distributions, both continuous and discrete.

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
