[Supplementary Material]

# A Normalizable distributions

*Proof of Proposition 1 (distributions close to normalizable sets are approximately normalizable).*

Let $T(x, y) = T^*(x, y) + T^-(x, y)$, where $T^*(x, y) = \underset{T(x,y):x\in\mathcal{S}}{\arg\min} ||T(X, y) - T(x, y)||_2$.

Then,

$$\mathbb{E}\left(\log\left(\int e^{\eta^\top T(X,y)}\,\mathrm{d}y\right)\right)^2 = \mathbb{E}\left(\log\left(\int e^{\eta^\top(T^*(X,y)+T^-(X,y))}\,\mathrm{d}y\right)\right)^2$$

$$\leq \mathbb{E}\left(\log\left(e^{\eta^\top\tilde{T}}\int e^{\eta^\top T^*(X,y)}\,\mathrm{d}y\right)\right)^2$$

for $\tilde{T} = \underset{T(X,y)}{\arg\max} ||\eta^\top T(X, y)||_2$,

$$\leq \mathbb{E}\left(\log\left(e^{\eta^\top\tilde{T}}\right)\right)^2$$

$$= (DB)^2 \qquad \square$$

# B Normalization and likelihood

## B.1 General bound

**Lemma 5.** *If $||\eta||_2 \leq \delta/R$, then $p_\eta(y|x)$ is $\delta$-approximately normalized about $\log\mu(\mathcal{Y})$.*

*Proof.* If $\int e^{\eta^\top T(X,y)}\,\mathrm{d}\mu(y) \geq \log\mu(\mathcal{Y})$,

$$\left(\log\int_\mathcal{Y} e^{\eta^\top T(X,y)}\,\mathrm{d}\mu(y) - \log\mu(\mathcal{Y})\right)^2 \leq \left(\log\int_\mathcal{Y} e^{||\eta||_2 R}\,\mathrm{d}\mu(y) - \log\mu(\mathcal{Y})\right)^2$$

$$= ||\eta||_2^2 R^2$$

$$\leq \delta^2$$

The case where $\int e^{\eta^\top T(X,y)}\,\mathrm{d}\mu(y) \leq \log\mu(\mathcal{Y})$ is analogous, instead replacing $\eta^\top T(x, y)$ with $-||\eta||_2 R$. The variance result follows from the fact that every log-partition is within $\delta$ of the mean. $\square$

*Proof of Theorem 2 (loss of likelihood is bounded in terms of distance from uniform).* Consider the likelihood evaluated at $\alpha\hat{\eta}$, where $\alpha = \delta/R||\hat{\eta}||_2$. We know that $0 \leq \alpha \leq 1$ (if $\delta > R\eta$, then the MLE already satisfying the normalizing constraint). Additionally, $p_{\alpha\hat{\eta}}(y|x)$ is $\delta$-approximately normalized. (Both follow from Lemma 5.)

Then,

$$\Delta_\ell = \frac{1}{n}\sum_i\left[(\hat{\eta}^\top T(x_i, y_i) - A(x_i, \hat{\eta})) - (\alpha\hat{\eta}^\top T(x_i, y_i) - A(x_i, \alpha\hat{\eta}))\right]$$

$$= \frac{1}{n}\sum_i\left[(1-\alpha)\hat{\eta}^\top T(x_i, y_i) - A(x_i, \hat{\eta}) + A(x_i, \alpha\hat{\eta})\right]$$

Because $A(x, \alpha\eta)$ is convex in $\alpha$,

$$A(x_i, \alpha\hat{\eta}) \leq (1-\alpha)A(x_i, \mathbf{0}) + \alpha A(x_i, \hat{\eta})$$

$$= (1-\alpha)\mu(\mathcal{Y}) + \alpha A(x_i, \hat{\eta})$$

Thus,

$$\Delta_\ell = \frac{1}{n}\sum_i \left[(1-\alpha)\hat\eta^\top T(x_i, y_i) - A(x_i, \hat\eta) + (1-\alpha)\log\mu(\mathcal{Y}) + \alpha A(x_i, \hat\eta)\right]$$

$$= (1-\alpha)\frac{1}{n}\sum_i \left[\hat\eta^\top T(x_i, y_i) - A(x_i, \hat\eta) + \log\mu(\mathcal{Y})\right]$$

$$= (1-\alpha)\frac{1}{n}\sum_i \left[\log p_\eta(y|x) - \log\mathrm{Unif}(y)\right]$$

$$\asymp (1-\alpha)\,\mathbb{E}\,\mathrm{KL}(p_\eta(\cdot|X) \,||\, \mathrm{Unif})$$

$$\leq \left(1 - \frac{\delta}{R||\hat\eta||_2}\right)\mathbb{E}\,\mathrm{KL}(p_\eta(\cdot|X) \,||\, \mathrm{Unif}) \qquad\qquad \square$$

## B.2   All-nonuniform bound

We make the following assumptions:

- Labels $y$ are discrete. That is, $\mathcal{Y} = \{1, 2, \ldots, k\}$ for some $k$.

- $x \in \mathcal{H}(d)$. That is, each $x$ is a $\{0, 1\}$ indicator vector drawn from the Boolean hypercube in $q$ dimensions.

- Joint feature vectors $T(x, y)$ are just the features of $x$ conjoined with the label $y$. Then it is possible to think of $\eta$ as a sequence of vectors, one per class, and we can write $\eta^\top T(x, y) = \eta_y^\top x$.

- As in the body text, let all MLE predictions be nonuniform, and in particular let each $\hat\eta_{y^*}^\top x - \hat\eta_y^\top x > c||\hat\eta||$ for $y \neq y^*$.

**Lemma 6.** *For a fixed $x$, the maximum covariance between any two features $x_i$ and $x_j$ under the model evaluated at some $\eta$ in the direction of the MLE:*

$$Cov[T(X, Y)_i, T(X, Y)_j | X = x] \leq 2(k-1)e^{-c\delta} \qquad (12)$$

*Proof.* If either $i$ or $j$ is not associated with the class $y$, or associated with a zero element of $x$, then the associated feature (and thus the covariance at $(i, j)$) is identically zero. Thus we assume that $i$ and $j$ are both associated with $y$ and correspond to nonzero elements of $x$.

$$\mathrm{Cov}[T_i, T_j | X = x] = \sum_y p_\eta(y|x) - p_\eta(y|x)^2$$

Suppose $y$ is the majority class. Then,

$$p_\eta(y|x) - p_\eta(y|x)^2 = \frac{e^{\eta_y^\top x}}{\sum_{y'} e^{\eta_{y'}^\top x}} - \frac{e^{2\eta_y^\top x}}{\left(\sum_{y'} e^{\eta_{y'}^\top x}\right)^2}$$

$$= \frac{e^{\eta_y^\top x}\left(\sum_{y'} e^{\eta_{y'}^\top x}\right) - e^{2\eta_y^\top x}}{\left(\sum_{y'} e^{\eta_{y'}^\top x}\right)^2}$$

$$\leq \frac{e^{\eta_y^\top x}\left(\sum_{y'} e^{\eta_{y'}^\top x}\right) - e^{2\eta_y^\top x}}{e^{2\eta_y^\top x}}$$

$$= \sum_{y' \neq y} e^{(\eta_{y'}' - \eta_y)^\top x}$$

$$\leq (k-1)e^{-c||\eta||}$$

Now suppose $y$ is not in the majority class. Then,

$$p_\eta(y|x) - p_\eta(y|x)^2 \le p(y|x)$$
$$= \frac{e^{\eta_y^\top x}}{\sum_{y'} e^{\eta_{y'}^\top x}}$$
$$\le e^{-c||\eta||}$$

Thus the covariance

$$\sum_y p_\eta(y|x) - p_\eta(y|x)^2 \le 2(k-1)e^{-c||\eta|||}$$

$\square$

**Lemma 7.** *Suppose $\eta = \beta\hat\eta$ for some $\beta < 1$. Then for a sequence of observations $(x_1, \ldots, x_n)$, under the model evaluated at $\xi$, the largest eigenvalue of the feature covariance matrix*

$$\frac{1}{n}\sum_i \left[ \mathbb{E}_\xi[TT^\top|X = x_i] - (\mathbb{E}_\theta[T|X = x_i])(\mathbb{E}_\xi[T|X = x_i])^\top \right] \tag{13}$$

*is at most*

$$q(k-1)e^{-c\beta||\hat\eta||} \tag{14}$$

*Proof.* From Lemma 6, each entry in the covariance matrix is at most $(k-1)e^{-c||\eta||} = (k-1)e^{-c\beta||\hat\eta||}$. At most $q$ features are nonzero active in any row of the matrix. Thus by Gershgorin's theorem, the maximum eigenvalue of each term in Equation 13 is $q(k-1)e^{-c\beta||\hat\eta||}$, which is also an upper bound on the sum. $\square$

*Proof of Proposition 3 (loss of likelihood goes as $e^{-\delta}$).* As before, let us choose $\hat\eta_\delta = \alpha\hat\eta$, with $\alpha = \delta/R||\hat\eta||_2$. We have already seen that this choice of parameter is normalizing.

Taking a second-order Taylor expansion about $\eta$, we have

$$\log p_{\hat\eta_\delta}(y|x) = \log p_\eta(y|x) + (\hat\eta_\delta - \hat\eta)^\top \nabla \log p_{\hat\eta}(y|x) + (\hat\eta_\delta - \hat\eta)^\top \nabla\nabla^\top \log p_\xi(y|x)(\hat\eta_\delta - \hat\eta)$$
$$= \log p_{\hat\eta}(y|x) + (\hat\eta_\delta - \hat\eta)^\top \nabla\nabla^\top \log p_\xi(y|x)(\hat\eta_\delta - \hat\eta)$$

where the first-order term vanishes because $\hat\eta$ is the MLE. It is a standard result for exponential families that the Hessian in the second-order term is just Equation 13. Thus we can write

$$\ge \log p_{\hat\eta}(y|x) - ||\hat\eta_\delta - \hat\eta||^2 q(k-1)e^{-c\beta||\eta||}$$
$$\ge \log p_{\hat\eta}(y|x) - (1-\alpha)^2||\hat\eta||^2 q(k-1)e^{-c\alpha||\eta||}$$
$$= \log p_{\hat\eta}(y|x) - (||\hat\eta|| - \delta/R)^2 q(k-1)e^{-c\delta/R}$$

The proposition follows. $\square$

## C Variance lower bound

Let

$$U_0 = \{\beta \in \mathbb{R}^{Kd} \colon \exists \tilde\beta \in \mathbb{R}^d, \beta_{kj} = \tilde\beta_j, \ 1 \le k \le K, \ 1 \le j \le d\}.$$

**Lemma 8.** *If* $\mathrm{span}\,(\mathcal{X}) = \mathbb{R}^d$, *then equivalence of natural parameters is characterized by*

$$\eta \sim \eta' \iff \eta - \eta' \in U_0.$$

*Proof.* For $x \in \mathcal{X}$, denote by $P_\eta(x) \in \Delta_K$ the distribution over $\mathcal{Y}$. Now, suppose that $\eta \sim \eta'$ and fix $x \in \mathcal{X}$. By the definition of equivalence, we have

$$\frac{P_\eta(x)_k}{P_\eta(x)_{k'}} = \frac{P_{\eta'}(x)_k}{P_{\eta'}(x)_{k'}},$$

which immediately implies

$$(\eta_k - \eta_{k'})^T x = (\eta'_k - \eta'_{k'})^T x,$$

whence

$$[(\eta_k - \eta'_k) - (\eta_{k'} - \eta'_{k'})]^T x = 0.$$

Since this holds for all $x \in \mathcal{X}$ and $\mathrm{span}(\mathcal{X}) = \mathbb{R}^d$, we get

$$\eta_k - \eta'_k = \eta_{k'} - \eta'_{k'}.$$

That is, if we define

$$\tilde{\beta}_j = \eta_{1j} - \eta'_{1j},$$

we get

$$\eta_{kj} - \eta'_{kj} = \eta_{1d} - \eta'_{1j} = \tilde{\beta}_j,$$

and $\eta - \eta' \in U_0$, as required.

Conversely, if $\eta - \eta' \in U_0$, choose an appropriate $\tilde{\beta}$. We then get

$$\eta_k^T x = (\eta')^T x + \tilde{\beta}^T x.$$

It follows that

$$A(\eta', x) = A(\eta, x) + \tilde{\beta}^T x,$$

so that

$$\eta^T T(k, x) - A(\eta, x) = (\eta')^T x + \tilde{\beta}^T x - \left[ A(\eta', x) + \tilde{\beta}^T x \right] = (\eta')^T x - A(\eta', x)$$

and the claim follows. $\qquad\square$

The key tool we use to prove the theorem reinterprets $V^*(\eta)$ as the norm of an orthogonal projection. We believe this may be of independent interest. To set it up, let $\mathcal{S} = L^2\left(Q, \mathbb{R}^D\right)$ be the Hilbert space of square-integrable functions with respect to the input distribution $p(x)$, define

$$w_j(x) = x_j - \mathbb{E}_{p(x)}\left[X_j\right]$$

and

$$\mathcal{C} = \mathrm{span}\left(w_j\right)_{1 \leq j \leq d}.$$

We then have

**Lemma 9.** *Let* $\tilde{A}(\eta, x) = A(\eta, x) - \mathbb{E}_{p(x)}\left[A(\eta, X)\right]$. *Then*

$$V^*(\eta) = \left\| \tilde{A}(\eta, \cdot) - \Pi_{\mathcal{C}} \tilde{A}(\eta, \cdot) \right\|_2^2.$$

The second key observation, which we again believe is of independent interest, is that under certain circumstances, we can completely replace the normalizer $A(\eta, \cdot)$ by $\max_{y \in \mathcal{Y}} \eta^T T(y, x)$. For this, we define

$$E_\infty(\eta)(x) = \max_k \eta^T T(k, x) = \max_k \eta_k^T x$$

and correspondingly let $\bar{E}_\infty(\eta) = \mathbb{E}_{p(x)}\left[E_\infty(\eta)(x)\right]$.

*Proof.* By Lemma 8, we have

$$V^*(\eta) = \inf_{\beta \in \mathbb{R}^d} \int_{\mathbb{R}^{Kd}} \left[ A(\eta, x) - \bar{A}(\eta) - \left(\beta^T x - \beta^T \mathbb{E}_{p(x)}\left[X\right]\right) \right]^2 \mathrm{d}p(x).$$

But now, we observe that this can be rewritten with the aid of the isomorphism $\mathbb{R}^d \simeq \mathcal{C}$ defined by the identity

$$\beta^T x - \beta^T \mathbb{E}_{p(x)}\left[X\right] = \sum_j \beta_j w_j(x)$$

to read

$$V^*(\eta) = \inf_{f \in \mathcal{C}} \int_{\mathbb{R}^d} \left[ A(\eta, x) - \bar{A}(\eta) - f \right]^2 \mathrm{d}p(x) = \left\| \tilde{A}(\eta, \cdot) - \Pi_{\mathcal{C}} \tilde{A}(\eta, \cdot) \right\|_2^2,$$

as required. $\qquad\square$

**Lemma 10.** *Suppose for each $x \in \mathcal{X}$, there is a unique $k^* = k^*(x)$ such that $k^*(x) = \arg\max_k \eta_k^T x$ and such that for $k \neq k^*$, $\eta_k^T x \leq \eta_{k^*}^T x - \Delta$ for some $\Delta > 0$. Then*

$$\sup_{x \in \mathcal{X}} \left| A(\eta, x) - \bar{A}(\eta) - \left[ E_\infty(\eta)(x) - \bar{E}_\infty(\eta) \right] \right| \leq K e^{-\Delta \alpha}.$$

*Proof.* Denote by $\tilde{E}_\infty$ the centered version of $E_\infty$. Using the identity $1 + t \leq e^t$, we immediately see that

$$E_\infty(\alpha\eta)(x) \leq A(\alpha\eta, x) = \alpha E_\infty(\eta)(x) + \log\left( 1 + \sum_{k \neq k^*(x)} e^{\left[ \eta_k^T x - E_\infty(\eta)(x) \right]} \right) \leq E_\infty(\alpha\eta)(x) + K e^{-\Delta \alpha}.$$

It follows that

$$\mathbb{E}_{p(x)} \left[ E_\infty(\alpha\eta)(X) \right] \leq \mathbb{E}_{p(x)} \left[ A(\alpha\eta, X) \right] \leq \mathbb{E}_{p(x)} \left[ E_\infty(\alpha\eta)(X) \right] + K e^{-\Delta \alpha}.$$

We thus have

$$-K e^{-\Delta \alpha} \leq \tilde{A}(\alpha\eta, x) - \tilde{E}_\infty(\alpha\eta)(x) \leq K e^{-\Delta \alpha}, \quad x \in \mathcal{X}.$$

The claim follows. $\qquad\square$

If we let

$$V_{\mathrm{E}}^*(\eta) = \inf_{\eta' \sim \eta} \mathrm{Var}_{p(x)} \left[ \tilde{E}_\infty(\eta', X) \right].$$

**Corollary 11.** *For $\alpha > \frac{\log 2K}{\Delta}$, we have*

$$V^*(\alpha\eta) \geq V_{\mathrm{E}}^*(\eta)\alpha^2 - (1 + V_{\mathrm{E}}^*(\eta))\alpha.$$

*Proof.* For this, observe first that if $\eta' \sim \eta$, then

$$\tilde{A}(\eta', x)^2 \geq \tilde{E}_\infty(\alpha\eta')(x)^2 - 2 \left| \tilde{E}_\infty(\alpha\eta')(x) \right| \left| \tilde{A}(\eta', x) - \tilde{E}_\infty(\eta')(x) \right|.$$

By linearity of $E_\infty(\eta')$ in its $\eta$ argument, and by Lemma 10, we therefore deduce

$$\tilde{A}(\eta', x)^2 \geq \tilde{E}_\infty(\eta')(x)^2 \alpha^2 - 2K e^{-\Delta \alpha} \left| \tilde{E}_\infty(\eta')(x) \right| \alpha.$$

Then using the inequality $\mathbb{E}_{p(x)} \left[ |f(X)| \right] \leq 1 + \mathrm{Var}_{p(x)} \left[ f(X) \right]$, valid for any $f \in L^2 \left( Q, \mathbb{R}^D \right)$ with $\mathbb{E}_{p(x)} [f] = 0$, we thus deduce

$$\mathrm{Var}_{p(x)} \left[ A(\alpha\eta', X) \right] \geq \mathrm{Var}_{p(x)} \left[ E_\infty(\eta')(X) \right] \alpha^2 - 2K e^{-\Delta \alpha} \left( 1 + \mathrm{Var}_{p(x)} \left[ E_\infty(\eta')(X) \right] \right) \alpha.$$

Taking the infimum over both sides, we get

$$V^*(\eta) \geq V_{\mathrm{E}}^*(\eta) - 2K e^{-\Delta \alpha} \left( 1 + V_{\mathrm{E}}^*(\eta) \right) \alpha.$$

$\qquad\square$

We are now prepared to give the explicit example. It is defined by $\eta_k = 0$ if $k > 2$ and

$$\eta_{1j} = \begin{cases} -a & \text{if } d = 1, \\ \frac{a}{d-1} & \text{o.w.} \end{cases} \tag{15}$$

and for all $j$,

$$\eta_{2j} = \frac{a}{d(d-1)}, \tag{16}$$

where

$$a = \sqrt{1 - \frac{1}{d}}.$$

For convenience, also define

$$b(x) = \sum_d x_d$$

and observe that

$$E_\infty(\eta)(x) = \begin{cases} \frac{ab(x)}{d(d-1)} & \text{if } x_j = 1, \\ \frac{ab(x)}{d-1} & \text{o.w.} \end{cases},$$

Our goal will be to prove that

$$1 \geq V_{\mathrm{E}}^*(\eta) \geq \frac{1}{32d(d-1)}.$$

The claim will then follow by the above corollary.

To see that $V_{\mathrm{E}}^*(\eta) \leq 1$, we simply note that

$$\max_k \left| \eta_k^T x \right| \leq a < 1,$$

whence $\mathrm{Var}_{p(x)} \left[ \eta^T x \right] \leq 1$ as well and we are done.

The other direction requires more work. To prove it, we first prove the following lemma

**Lemma 12.** *With $\eta$ defined as in (15)-(16), we have*

$$\inf_{\eta' \sim \eta} \mathbb{E}_{p(x)} \left[ E_\infty(\eta')(X)^2 \right] \geq \frac{1}{16d(d-1)}.$$

*Proof.* Suppose $\eta_k - \eta_k' = \beta \in \mathbb{R}^d$. We can then write

$$\inf_{\eta' \sim \eta} \mathbb{E}_{p(x)} \left[ E_\infty(\eta')(X)^2 \right] = \inf_{\beta \in \mathbb{R}^d} \frac{1}{2^d} \sum_{x \in \mathcal{H}} \sum_{x \in \mathcal{H}} \left[ E_\infty(\eta)(x) - \beta^T x \right]^2$$

and we therefore define

$$\mathcal{L}(\beta) = \sum_{x \in \mathcal{H}} \sum_{x \in \mathcal{H}} \left[ E_\infty(\eta)(x) - \beta^T x \right]^2$$

$$= \sum_{x : x_1 = 0} \left[ \left( \beta_1 + \beta^T x - \frac{a}{d(d-1)} \right)^2 + \left( \frac{ab(x)}{d-1} - \beta^T x \right)^2 \right],$$

noting that

$$\inf \mathcal{L} = 2^d \cdot \inf_{\eta' \sim \eta} \mathbb{E}_{p(x)} \left[ E_\infty(\eta')(X)^2 \right].$$

We therefore need to prove

$$\mathcal{L} \geq \frac{2^{d-4}}{d(d-1)}.$$

Holding $\beta_{2:d}$ fixed, we note that the optimal setting of $\beta_1$ is given by

$$\beta_1 = -\frac{1}{2} \sum_{j \geq 2} \beta_j + \frac{a}{d(d-1)}.$$

We can therefore work with the objective

$$\mathcal{L}(\beta) = \sum_{x : x_1 = 0} \left[ \frac{(\beta^T x - \beta^T x^\neg)^2}{4} + \left( \frac{ab(x)}{d-1} - \beta^T x \right)^2 \right],$$

where we have defined

$$x_j^\neg = \begin{cases} 0 & \text{if } j = 1, \\ 1 - x_j & \text{o.w.} \end{cases}$$

Grouping into $\{x, \ x^\neg\}$ pairs, we end up with

$$\mathcal{L}(\beta_{2:d}) = \sum_{x : x_1 = x_2 = 0} \left[ \frac{(\beta^T x - \beta^T x^\neg)^2}{2} + \left( \frac{ab(x)}{d-1} - \beta^T x \right)^2 + \left( \frac{ab(x^\neg)}{d-1} - \beta^T x^\neg \right)^2 \right]$$

Now, supposing $b(x) \leq \frac{d-1}{2} - \frac{3}{2}$ or $b(x) \geq \frac{D-1}{2} + \frac{3}{2}$, we have

$$|b(x^\neg) - b(x)| = |d - 1 - 2b(x)| \geq 3.$$

We will bound the terms that satisfy this property. Indeed, supposing we fix such an $x$, at least one of the following must be true: either

$$\max\left(\left(\frac{ab(x)}{d-1} - \beta^T x\right)^2, \left(\frac{ab(x^\neg)}{d-1} - \beta^T x^\neg\right)^2\right) \geq \frac{a^2}{(d-1)^2},$$

or

$$\left(\beta^T x - \beta^T x^\neg\right)^2 \geq \frac{a^2}{(d-1)^2}.$$

Indeed, suppose the first condition does not hold. Then necessarily

$$\left|\frac{ab(x)}{d-1} - \beta^T x\right| < \frac{a}{d-1}$$

and

$$\left|\frac{ab(x^\neg)}{d-1} - \beta^T x^\neg\right| < \frac{a}{d-1},$$

so that

$$\frac{a(b(x)-1)}{d-1} \leq \beta^T x \leq \frac{a(b(x)+1)}{d-1}$$

and

$$\frac{a(b(x^\neg)-1)}{d-1} \leq \beta^T x \leq \frac{a(b(x^\neg)+1)}{d-1}.$$

Now, if $b(x) \geq b(x^\neg) + 3$, this immediately implies

$$\beta^T x - \beta^T x^\neg \geq \frac{a}{d-1}$$

and, symmetrically, if $b(x^\neg) \geq b(x) + 3$, we get

$$\beta^T x^\neg - \beta^T x \geq \frac{a}{d-1}.$$

Either way, the second inequality holds, whence the claim. Since there are at least $2^{d-1} - \frac{3 \cdot 2^d}{\sqrt{\frac{3d}{2}+1}} \geq 2^{d-2}$ choices of $x$ satisfying the requirements of our line of reasoning, we get $2^{d-3}$ pairs, whence

$$\mathcal{L}(\beta_{2:d}) \geq \frac{2^{d-4}a^2}{(d-1)^2} = \frac{2^{d-4}}{d(d-1)},$$

as claimed. □

We can apply this lemma to derive a variance bound, viz.

**Lemma 13.** *With $\eta$ as in (15)-(16), we have*

$$V_E^*(\eta) \geq \frac{1}{32d(d-1)}.$$

*Proof.* For this, observe that, with $\eta'$ being the value corresponding to $\eta'_k - \eta_k = \beta$, we have

$$V_E^*(\eta) = \inf_\beta \frac{1}{2^d} \sum_{x \in \mathcal{H}} \tilde{E}_\infty(\eta')(x)^2 \geq \inf_\beta \frac{1}{2^d} \sum_{x \in \mathcal{H}: \, x_1 = 1} \tilde{E}_\infty(\eta')(x)^2.$$

Applying the previous result to the $(D-1)$-dimensional hypercube on which $x_1 = 1$, we deduce

$$V_E^*(\eta) \geq \frac{1}{2} \cdot \frac{1}{16(d-1)(d-2)} = \frac{1}{32(d-1)(d-2)} \geq \frac{1}{32d(d-1)}.$$

□

*Proof of Theorem 4 from Lemma 13.* Putting everything together, we see first that

$$V^*(\alpha\eta) \geq V_{\mathrm{E}}^*(\eta)\alpha^2 - 4e^{-\Delta\alpha}\alpha,$$

where $\Delta = \frac{\sqrt{1-\frac{1}{d}}}{2(d-1)}$. But then this implies

$$V^*(\alpha\eta) \geq \frac{\alpha^2}{32d(d-1)} - 4e^{-\Delta\alpha}\alpha.$$

On the other hand, $||\eta||_2^2 \leq 2$, so $\alpha^2 = \frac{||\alpha\eta||_2^2}{||\eta||_2^2} \geq \frac{||\alpha\eta||_2^2}{2}$, whence

$$V^*(\alpha\eta) \geq \frac{||\alpha\eta||_2^2}{64d(d-1)} - 4e^{-\frac{\sqrt{1-\frac{1}{d}}||\alpha\eta||_2}{2(d-1)}} ||\alpha\eta||_2,$$

which is the desired result. $\qquad\square$