[Reviews · NeurIPS 2015]

Submitted by Assigned_Reviewer_1

This paper addresses the use of self-normalising distributions to address the problem of expensive normalisation for problems with many classes. Using log-linear as reasonable class of models the paper first introduces definitions of closeness to self-normalisability and then goes on to derive novel results regarding the the error incurred as a function of closeness.

The paper addresses a timely topic, is clear with a good mix of theoretical results and more informal intuitive reasoning. The characterisation of distributions in terms of their entropy makes a lot of sense. The Experiments Section is short but supports the points of the paper well. A bit more detail should be given on set-up and details of study.
Summary: This paper addresses the use of self-normalising distributions to address the problem of expensive normalisation for problems with many classes. Using log-linear as reasonable class of models the paper first introduces definitions of closeness to self-normalisability and then goes on to derive novel results regarding the the error incurred as a function of closeness.

Submitted by Assigned_Reviewer_2

I enjoyed this paper, which defined some useful theoretical machinery to help us to understand self-normalizing models, and then employed this machinery to yield some interesting results. I expect that the framework will support further investigation, perhaps on the open questions defined at the end of the paper; I'm particularly interested in knowing more about the tightness of these bounds.

For the practitioner, it would be helpful to have a little more explanation about the properties of these bounds, as well as some intuition about quantities like R, which here defines the maximum L2 norm of the parameter vector -- for example, is this value typically bounded in settings of practical interest?

I'm not aware of other work on this topic, except the NAACL paper by Andreas and Klein, and this submission goes well further than that paper. Drawing further connections to work on NCE would be of value, since this a well-known alternative to self-normalized models.

The paper packs in a lot of content in eight pages. The only downside is that this makes it a little difficult to read in places -- I found myself frequently paging back to remember things.
Summary: This paper adds useful theoretical results to our understanding of self-normalizing log-linear models.

Submitted by Assigned_Reviewer_3

This paper provides a theoretical analysis of self-normalized log-linear models, which are log-linear models trained with a penalty to keep the partition function close to one, so that scores can be directly interpreted as probabilities. These models are being used in the NLP community when the output space is large. The authors provide 1) a simple geometric characterization of exact and approximate self-normalization 2) a natural framework to characterize the tradeoff between fitting the data and keeping the log-partition function close to zero 3) some results showing that distributions that are either close to uniform or close to degenerate can be self-normalized (without losing too much in terms of likelihood).

The paper is well written, it provides intuitive explanations for the results, examples and figures. I would have liked to see more discussion for the results at the end of section 4 (looks like it was finished in a hurry).

As far as I know, the results are novel. The theoretical results shed some light on the role of self-normalization. The framework developed for the analysis is of (independent) interest and could be used by other researchers, e.g. to obtain deeper characterizations of the tradeoffs involved. This work could be of interest to people working on models where dealing with the intractable log-normalizer is the key computational bottleneck (on energy based models, noise constrastive estimation, etc). One limitation is that the results presented don't directly lead to new algorithms, and it's not clear to me if they are useful to inform the work of a practitioner (e.g, how to set parameters or design models).

I am also not sure about the significance and the possible implications of Theorem 4. How does the model fit (data likelihood) enter the picture? Can you say something about small perturbations of \eta_0?

More details should be given about the experimental evaluations. Currently, there is not enough information to replicate the results (what is the X, Y, features used etc.)

in proposition 1, should it be ||\eta||_2 <= B ?
Summary: Interesting first step towards analyzing self-normalized log-linear models. The paper provides a nice geometrical interpretation and a formal framework to characterize the tradeoff between model fit and the self-normalization property. Within this framework, the authors provide some lower and upper bounds on what can and cannot be achieved with self-normalization. This is an interesting first step, although many questions remain open.

Author Feedback
Author rebuttal: We thank the reviewers for their helpful feedback, and will be sure to fix all typos noted in the reviews. Additionally,

(1) R2 asks about NCE as an alternative to self-normalization. In fact, the two are complementary: NCE allows us to avoid computing normalizers during training, but may still require normalizer computation at test time. On the other hand, self-normalization requires computation during training, but allows computation to be skipped at test time. It is possible to do both simultaneously (as in Vaswani et al., 2013) in which case the analysis in this paper applies directly.

(2) R3 requests additional information about the experimental setup. We will provide more detail (training set sizes, numbers of features and classes, etc.) in any final version of the paper.